# Integrating Mobile Thermal Energy Storage (M-TES) in the City of Surrey's District Energy Network: A Techno-Economic Analysis

Maha Shehadeh [1], Emily Kwok [2], Jason Owen [2] and Majid Bahrami [1,*]

1 Pacific Institute for Climate Solutions and Laboratory for Alternative Energy Conversion (LAEC), School of Mechatronic Systems Engineering & School of Sustainable Energy Engineering, Simon Fraser University, 250-13450 102 Ave, Surrey, BC V3T 0A3, Canada; mahas@sfu.ca

2 Engineering Department, City of Surrey, Surrey City Hall 13450—104 Avenue Surrey, BC V3T 1V8, Canada; Emily.Kwok@surrey.ca (E.K.); jowen@surrey.ca (J.O.)

* Correspondence: mbahrami@sfu.ca

**Abstract:** The City of Surrey in British Columbia, Canada has recently launched a district energy network (DEN) to supply residential and commercial buildings in the Surrey Centre area with hot water for space and domestic hot water heating. The network runs on natural gas boilers and geothermal exchange. However, the City plans to transition to low-carbon energy sources and envisions the DEN as a key development in reaching its greenhouse gas emissions (GHG) reduction targets in the building sector. Harvesting and utilizing waste heat from industrial sites using a mobile thermal energy storage (M-TES) is one of the attractive alternative energy sources that Surrey is considering. In this study, a techno-economic analysis (TEA) was conducted to determine the energy storage density (ESD) of the proposed M-TES technology, costs, and the emission reduction potential of integrating waste heat into Surrey's DEN. Three transportation methods were considered to determine the most cost-effective and low-carbon option(s) to transfer heat from industrial waste heat locations at various distances (15 km, 30 km, 45 km) to district energy networks, including: (i) a diesel truck; (ii) a renewable natural gas-powered (RNG) truck, and (iii) an electric truck. To evaluate the effectiveness of M-TES, the cost of emission reduction ($/tCO2e avoided) is compared with business as usual (BAU), which is using a natural gas boiler only. Another comparison was made with other low carbon energy sources that the city is considering, such as RNG/biomass boiler, sewer heat recovery, electric boiler, and solar thermal. The minimum system-level ESD required to makes M-TES competitive when compared to other low carbon energy sources was 0.4 MJ/kg.

**Keywords:** district energy network; mobile thermal energy storage (M-TES); industrial waste heat; techno-economic analysis; and GHG emission reduction; low carbon energy

## 1. Introduction

Buildings consume more than 40% of the total primary energy in developed countries, with approximately 70% of the buildings' energy demand being for space heating and domestic hot water [1]. Due to Canada's cold climate and its heavy reliance on fossil fuels for heating [2], cutting the greenhouse gas emissions (GHG) due to space and water heating is crucial to meet the country's national GHG emissions targets and international commitments under the Paris Climate Agreement [3]. The City of Surrey, in British Columbia, is one of the fastest-growing cities in Canada and has recently launched a district energy network (DEN) to supply hot water for more than 250,800 m$^2$ [4] of various residential and commercial buildings in the Surrey Centre area for space and domestic hot water heating. The network currently runs on natural gas boilers and geothermal exchange with highly insulated pipes to move hot water to various buildings, and exchanges energy using heat exchangers located at the customer side. Integrating renewable and low carbon

energy sources in district heating and cooling is vital to meet rising urban energy needs [5]. Surrey aims to improve energy efficiency, reduce GHG emissions, increase resilience and provide competitive pricing by using the DEN as a key development in reaching its GHG reduction targets in the building sector [6]. Low carbon Surrey DEN will be a collective energy system that employs multiple renewable energy technologies, e.g., biomass boilers, solar thermal and waste heat recovery. Using renewables and low-energy sources in DENs is a strategy that many communities are following to decarbonize heat productions; for example, (i) Aarhus, Denmark: electric boiler and heat pump for district heating [7], (ii) London Olympic Park, Great Britain: biomass boilers [8], (iii) Ulm, Germany: replacement of fossil fuel plants with biomass plants [9]. Other examples of using low-carbon energy sources in district heating and cooling can be found in the IRENA case studies report [10].

Utilizing waste heat from industrial sites is a potential technology that Surrey is considering as a possible low carbon energy source. In this paper, a techno-economic analysis (TEA) has been performed to determine the potential emission reduction of integrating industrial waste heat (IWH) into Surrey's DEN using a mobile thermal energy storage (M-TES), to capture, store and move excess heat from industrial locations to the Surrey district heating network. M-TES consists of a truck and thermal energy storage system.

One way to connect IWH and DEN is the use of pipelines, e.g., one Chinese city, Anshan, aims to limit the use of heavily polluting coal by a projected 1.2 million tons per year through connecting its district energy networks to capture waste heat from a local steel plant [11]. However, the cost of pipelines and the complexity of the system significantly increases when waste heat sources are in multiple areas and away from the DEN. Pipeline connections require a huge upfront cost, which limits the collection of waste heat from multiple locations to multiple DENs. On the other hand, storing and moving thermal energy in a truck using M-TES is a flexible option with a lower upfront cost [12]. Moreover, due to the Fraser river in Surrey district energy network case study, the pipeline option is not possible.

The TEA presented here examines scenarios for integrating industrial waste heat in terms of (i) locations (distance); (ii) transportation modes; and (iii) the energy storage densities of the targeted M-TES technologies. The goal is to assess the feasibility, cost-effectiveness, and overall environmental impact. The TEA also helps in estimating GHG emission savings in $tCO_{2e}/MWh$, estimates the useful heat from M-TES [MWh/year], and establishes target prices [$/MWh, $/$tCO_2$e avoided] that makes the proposed M-TES system economically viable.

## 2. Overview

The City of Surrey has committed to achieving net-zero emissions by 2050. In 2007, buildings in Surrey emitted 911,000 $tCO_{2e}$, more than 40% of the City's total emissions [13]. The Surrey City Energy (SCE) is part of Surrey's plan to reduce emissions from buildings. It is projected for the network to service more than 1,600,000 $m^2$ of built area by 2040, reducing the per unit area GHG emissions ($kgCO_{2e}/m^2$) from its service area by more than 70% compared to the 2007 baseline [13]. The newly built facility uses high-efficiency natural gas boilers. The City plans to connect more than 20 high-density buildings in Surrey's City Centre to a downtown DEN which will run on a portfolio of renewable energy sources, including geothermal exchange, biomass, waste heat, and sewer heat recovery [14].

In collaboration with the City of Surrey, our team has developed a model for thermal energy storage (TES), short-term storage, using a water tank for peak shaving, improving DEN efficiency, and reducing capital cost, as well as serving as a connection point between the proposed M-TES and the DEN [15]. To supply low-carbon heat to SCE cost-effectively and energy-efficiently, a new M-TES has been proposed to harvest and transfer freely available waste heat from industrial sites, where the low-grade energy (sources with temperature less than 90 °C) is released to the ambient unused, to Surrey's DEN. The M-TES system should be competitive with other available low-carbon sources, see Figure 1. Estimated levelized costs for other low carbon sources in this study were given by Surrey's DEN.

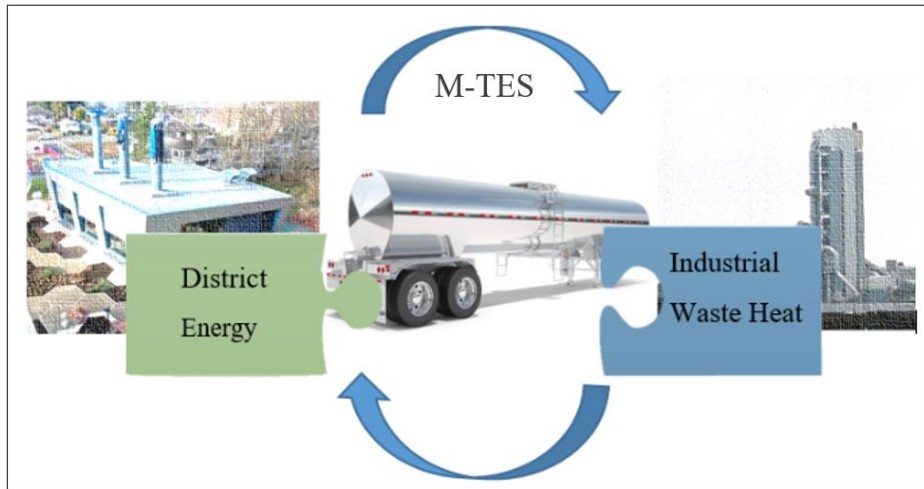

**Figure 1.** The concept design of mobile thermal energy storage (M-TES), the missing link, to move industrial waste heat to the District Energy Network.

The system energy storage density (ESD) of M-TES is a key performance indicator in determining M-TES feasibility and effectiveness. The system-level ESD is a function of the storage material ESD and the system efficiency in extracting and delivering the stored heat from the storage material. M-TES systems that are based on phase change materials (PCM) may not be suitable for integration into SCEs mainly due to their low ESD~0.25 MJ/kg [16]. Moreover, they require large, embedded heat exchangers to travel with the M-TES container, which further lowers the system-level ESD and increases transportation costs. Their low thermal conductivity is another challenge that increases the "charging/discharging" process and waste heat source temperature [17]. The economic and environmental feasibility of PCM M-TES systems has been studied in [12]. There is a need for the development of compatible thermal energy storage with high ESD and charge/discharge temperatures suitable for the industrial waste heat sources available for SCE. A comparison between the material-based ESD and the system level ESD for heat and cold storage extracted from the literature shows a wide range of ESD (0.260–1.603 GJ m$^{-3}$) [18]. Thermochemical materials (TCM) are promising candidates [19]. TCM storage such as LiBr/H$_2$O solution is based on the application of reversible chemical reactions. They are well-matched to our M-TES requirements due to their stable reversibility and high reaction enthalpies, resulting in high ESD. TCM solutions can be transported in tanks and can be readily pumped in and out without the need to transport heat exchangers. System-level ESD is always less than the material level ESD depending on the bulkiness and inefficiency of the system. The minimum system-level ESD that is required to make M-TES economically feasible is determined in this work.

### 2.1. M-TES Demand Estimation

The SCE supplies hot water to several buildings, many of which are high rises, in the downtown of Surrey Centre area and within a 1 km radius of the DEN. The City's forecast of energy demand is reproduced in Table 1 [4].

**Table 1.** The Surrey City Energy projected annual demand.

|  | 2022 | 2024 | 2035 |
|---|---|---|---|
| Annual energy demand, MWh | 56,600 | 70,000 | 130,300 |
| M-TES 7% target, MWh | 3900 | 4900 | 9100 |

The SCE has a GHG intensity target of 0.07 $tCO_{2e}$/MWh of heat sales [20]. Based on this target, the required energy from low carbon energy sources is illustrated in Figure 2. M-TES can be used in the decarbonization of the SCE harvesting and transferring industrial waste heat. The initial target is to supply at least 7% per M-TES truck of total demand in 2022, see Table 1.

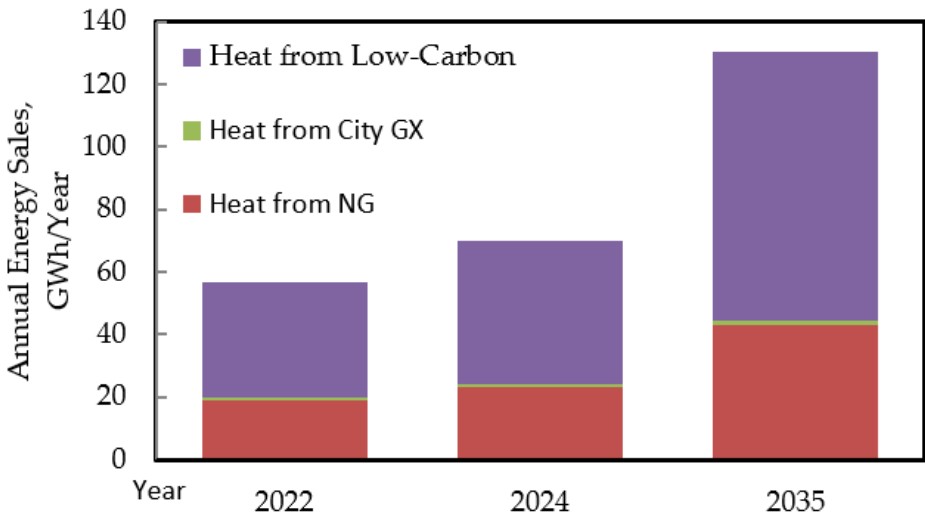

**Figure 2.** Surrey City Energy's projected annual demand and target energy sources.

To supply 7% of the total energy sales indicated in Figure 3, the system ESD should be around 0.7 MJ/kg assuming a 15 km distance radius between IWH location and DEN and one M-TES truck with trip schedule, as shown in Table 2.

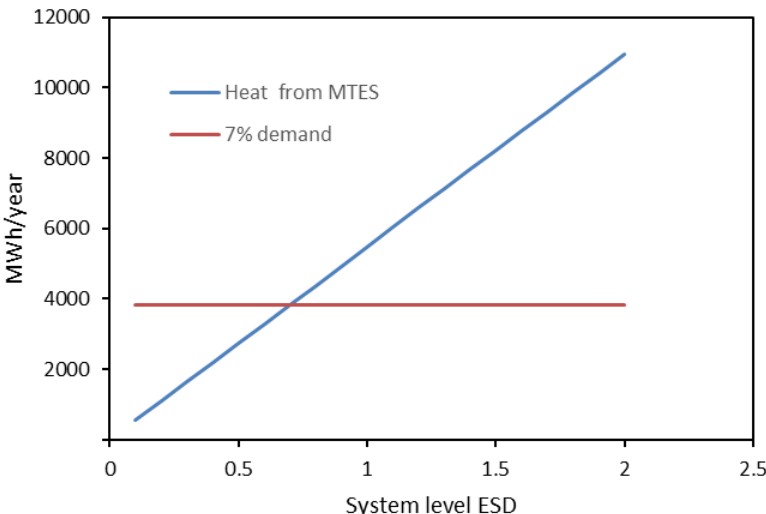

**Figure 3.** M-TES system energy storage density (ESD) required to achieve 7% of the City of Surrey's 2022 target.

**Table 2.** Parameters used in creating comparison graphs.

| Transportation Mode Options | RNG Truck<br>Diesel Truck<br>EV Truck |
|---|---|
| Distance (DEN to IWH) | D1 (15 km)<br>D2 (30 km)<br>D3 (45 km) |
| System level ESD | 0.1 MJ/kg–2.0 MJ/kg |
| Number of trips/day | 6 Trips |
| Number of days/year | 360 Days |
| Other low carbon heat sources. Cost is based on Surrey DEN given information | Solar, Sewer heat recovery, RNG, Biomass, Electric boiler |

*2.2. Industrial Waste Heat Identification*

Nearly 50% of industrial energy in Canada is lost as waste heat [21]. To increase energy efficiency, industrial heat recovery is a logical way to take advantage of the available waste heat and turn it into a usable energy source for local communities. There are different ways to capture waste heat energy from industrial processes and there are mainly two basic applications: (i) recycling waste heat back into the processes onsite and (ii) transferring the waste heat for use off-site, which is the focus for the M-TES technology. In our study, there is a need to transfer IWH because none of the industrial locations is close to the DEN or within the 1 km radius of the network. The possible location of available IWH is a cement plant within a 20 km radius. Cement production in particular has the most potential for regional heat recovery by industry [22]. Other IWH locations are also available within the Fraser River industrial area, with a similar proximity.

The distance from IWH locations to SCE is a crucial factor in determining the levelized cost and GHG emission reduction using M-TES. The distance range that is used in this study is 15 km–45 km, listed in Table 2. Within this range, there are multiple IWHs available [23].

**3. Methodology**

The economic evaluation of the M-TES system was conducted based on the cost of supplying heat. To identify the key parameters, the cost breakdown was analyzed. A parametric study was also conducted to understand the impacts of the following key parameters: (i) distance from IWH, (ii) ESD of the M-TES, and (iii) transportation mode, where electric, diesel, and renewable natural gas (RNG) trucks are considered. The price of IWH delivered by M-TES and the corresponding avoided emissions were compared to the BAU scenario with the existing natural gas boilers, as well as other low-carbon energy sources to determine the comparative advantage and potential of M-TES.

*3.1. Levelized Cost Model*

To objectively compare different storage technologies from an economical point of view, the Levelized Costs of M-TES or (LCM) has been introduced. Levelized costs are the ratio of the total lifetime expenses versus total expected outputs, expressed in terms of the present value equivalent [24,25]. It is used as a benchmarking tool to assess the cost-effectiveness of different energy generation technologies and has been broadly used for the evaluation of power generation costs. For example, IRENA estimated power generation costs of different technologies around the world between 2010 and 2019 [26,27]

Many parameters need to be considered when calculating LCM, i.e., initial cost of investment, maintenance and operations cost, fuel cost, the total output of the heat provided, and the life of the system.

The cost of supplying 1 MWh of non-payable IWH is calculated based on the capital cost, as well as the system running costs. The capital cost of M-TES, which includes the costs of a container, thermochemical solution, pumps, tanks, and two heat exchangers, is

estimated at $CAD 100,000 plus the cost of a medium-duty truck. Interest and depreciation costs were ignored. Cost estimation of liquid solution pumps, tanks, and heat exchangers can be found in the following references [28,29]. Moreover, [30] provides a reasonable estimate of capital and operation costs, as well as for GHG emission factors for the three proposed modes of transportation: (i) diesel truck, (ii) RNG truck, and (iii) electric truck, see Table 3. Transportation cost varies with transport, mode, and travelling distance. With the assumptions used in the analysis, see Table 4, the transport distance, modes, ESD, and schedules were compared against the SCE-levelized cost of other low-carbon heat sources.

**Table 3.** The assumptions used in the analysis.

| Surrey DEN Target GHG Intensity | 0.07 tCO$_2$e/MWh |
|---|---|
| Natural gas boiler efficiency, $\eta_{NG}$<br>M-TES system efficiency, $\eta_{M\text{-}TES}$ | 95%<br>90% |
| Natural gas fuel GHG intensity, $GHG_{NG}$ | 180 kgCO$_2$e/MWh |
| Constant heat supply from a vertical geothermal exchange system attached to DEN | 1100 MWh |
| Emission factors and transportation methodologies and guides | From [30,31] |
| System level energy storage density of M-TES | (0.1 MJ/kg–2 MJ/kg) |
| M-TES system lifetime | 12 years |
| Tank size/trip | 10,000 kg |

**Table 4.** The summary of truck modes and associated costs.

| | Truck Mode | | |
|---|---|---|---|
| | **RNG Truck** | **EV Truck** | **Diesel Truck** |
| System capital cost estimate CAD$ | 470,000 | 570,000 | 400,000 |
| Fuel price (CAD$/L diesel equivalent) | 0.47 * | 0.32 | 1.4 |
| Other transportation cost (OTC), e.g., insurance + maintenance + driver rate (CAD$/km) [32,33] | 1.4 | 1.4 | 1.4 |

* discounted price from British Columbia for low-carbon fuel standards (LCFS) trucks.

### 3.2. Techno-Economic Analysis (TEA)

The levelized cost of M-TES (LCM) in ($/MWh) can be estimated as [34,35]:

$$LCM = \left( CC + \left( \frac{TC}{year} \times SLT \right) \right) \div (TH \times SLT) \qquad (1)$$

where *CC* is the capital cost, *TC* is total transportation cost per year, *SLT* is the system lifetime, and *TH* is the total heat provided by the system over one year. To simplify the cost calculations, carbon tax and interest accumulation were ignored.

The total heat from M-TES (TH) in one year (MWh/year) can be calculated as:

$$TH = ESD \times TW \times N \times \eta \times ODPY \qquad (2)$$

where *ESD* is energy storage density, *TW* is Total material weight, *N* is the number of trips per day, *η* is M-TES system efficiency, and *ODPY* is the number of operating days per year, respectively.

The total GHG emitted from the system in one year, GHG $_{Transportation}$ in (tCO$_2$e/year), will be:

$$GHG_{Transportation} = 2\,D \times N \times ODPY \times CO_{2e} \text{ factor} \qquad (3)$$

where D is the distance between IWH and DEN.

The transportation cost factor (TCF) in ($/km) can be calculated as:

$$\text{TCF} = \text{OTC} + (\eta_{\text{Fuel}} \times \text{FP}) \tag{4}$$

where OTC is other transportation costs (insurance, maintenance, and driver rate) in $ per kilometer, $\eta_{\text{Fuel}}$ is the fuel efficiency in liters per 100 km, and FP is the fuel price in dollar per liter, respectively.

The total transportation cost, *TC*, in $/year, will be:

$$TC = 2D \times N \times ODPY \times TCF \tag{5}$$

where the transportation cost (TC) is in $ CAD dollar, *D* is the distance between IWH and DEN, *N* is number of trips per day, *ODPY* is the number of operating days per year, *TCF* is the transportation cost factor [32,33].

The GHG$_{\text{Avoided}}$ is the avoided GHG, when using M-TES instead of natural gas (NG) boilers in tCO$_2$e/year. Thus, we have to find out how much a natural gas boiler going to emit GHG when producing the same amount of heat that M-TES is bringing in, this is the first part of the equation. Then, the GHG that is emitted due to M-TES is subtracted and can be calculated as:

$$\text{GHG}_{\text{Avoided}} = ((\text{TH}_{\text{M-TES}} \times \eta_{\text{NG}}) \times (\text{GHG}_{\text{NG}})) - \text{GHG}_{\text{Transportation}} \tag{6}$$

where TH$_{\text{M-TES}}$ is the total heat from M-TES in one year (MWh/year), $\eta_{\text{NG}}$ is the efficiency of the boilers, and GHG$_{\text{NG}}$ is the GHG intensity when using NG boilers. GHG intensity is assumed as constant: 180 kg/MWh, and GHG$_{\text{Transportation}}$ is the GHG from transportation.

## 4. Results

Figure 4 shows the cost of thermal energy delivered to DEN using M-TES transportation modes and various system level energy storage densities compared to the levelized cost of other energy sources in $/MWh. Biomass boiler uses fuel from the biofuel facility with levelized cost of $70/MWh. To compete with biomass boiler, a diesel truck M-TES system, will need to have at least 0.4 MJ/kg ESD. However, an electric truck or RNG truck will need a system-level ESD of at least 0.3 MJ/kg. Similarly, Figure 5 shows the cost of the avoided emissions ($/tCO$_{2e}$ Avoided) for various transportation modes and ESDs. Both Figures 4 and 5 are for a transport distance of 15 km between the IWH location and the DEN of the City of Surrey. The cut-off system level ESD to make M-TES competitive is 0.3 MJ/kg. However, the thermal energy demand of the DEN should be considered in the analysis. The cut-off system-level ESD for M-TES technology to meet the target 7% of the total demand with a cost competitive with other low-carbon sources, in terms of both the levelized cost and the cost of avoided emissions, is estimated at 0.7 MJ/kg.

Figure 6 illustrates the cost ($/MWh) for the considered truck modes with variable distances between IWH source and DEN. Since diesel has the highest $/MWh, it was further compared with other low-carbon sources in Figure 7. The cost for other low-carbon resources is based on the Surrey DEN analysis.

Similarly, Figure 8 shows the cost of avoided emissions ($/tCO$_{2e}$ Avoided) for the considered truck modes with variable distances between the IWH source and the DEN. Since diesel has the highest $/tCO$_{2e}$, it was further compared with other low carbon sources in Figure 9.

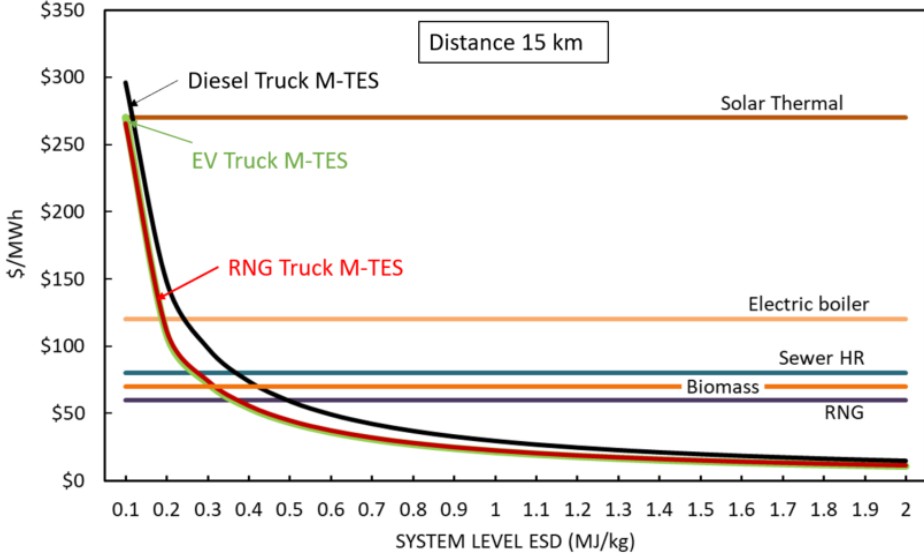

**Figure 4.** M-TES $/MWh with different ESD and transportation modes with a 15 km distance between the industrial waste heat and district energy network.

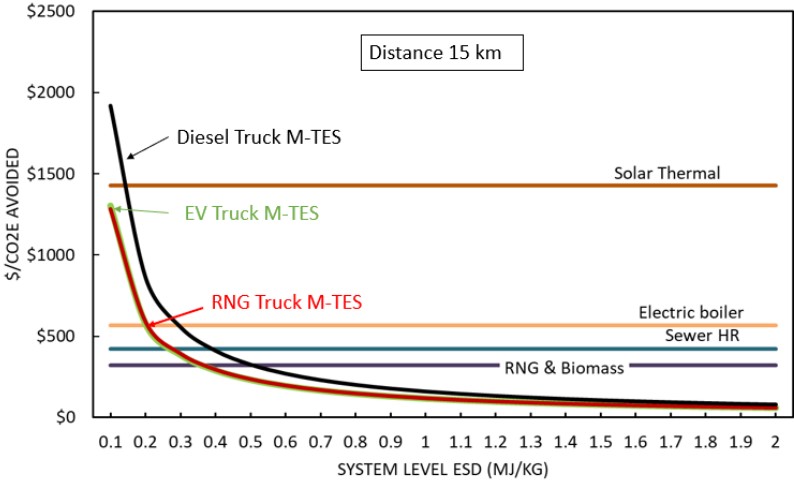

**Figure 5.** The avoided $/tCO$_2$e with different System ESD and transportation modes with 15 km distance between the IWS and DEN.

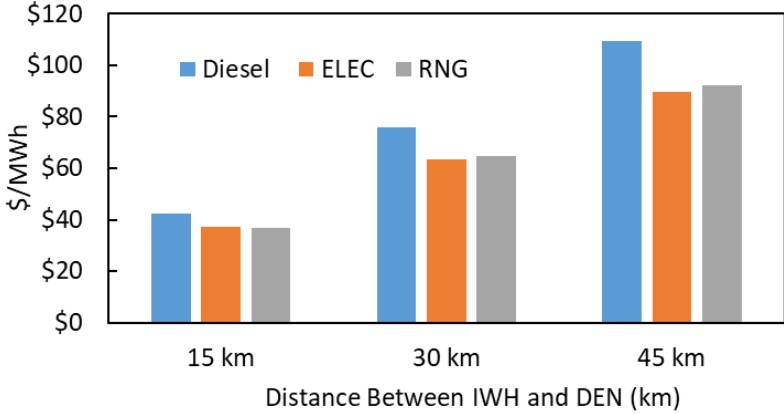

**Figure 6.** The M-TES levelized cost with different IWH locations and truck modes.

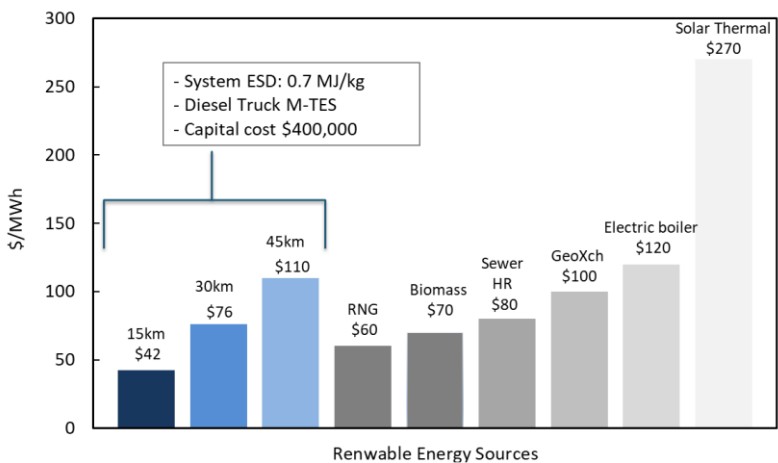

**Figure 7.** The levelized cost of diesel truck M-TES ($/MWh) varies with a distance and system ESD of 0.7 MJ/kg and other Surrey DEN low-carbon sources cost estimation.

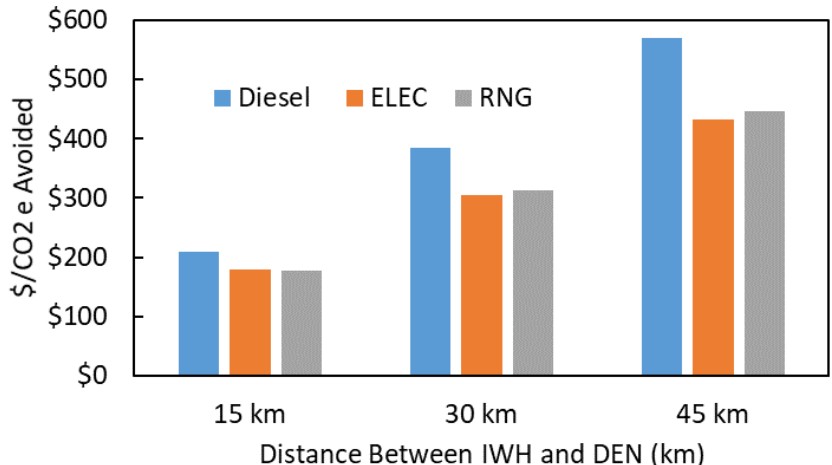

**Figure 8.** The cost of avoided $CO_2e$ with different IWH locations and truck modes.

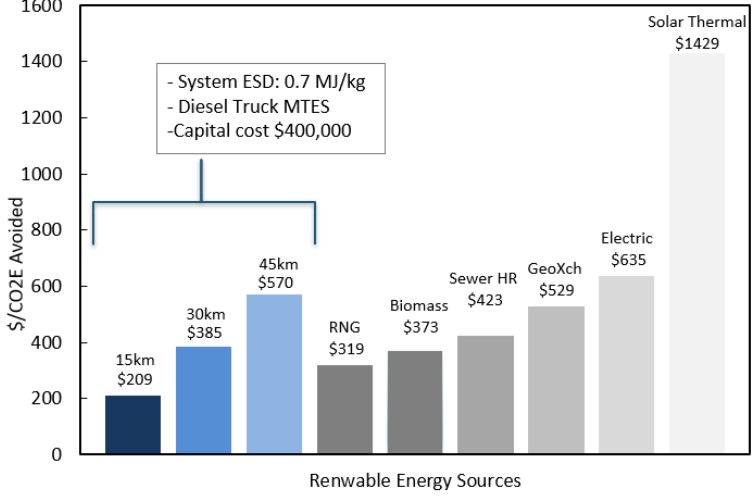

**Figure 9.** The avoided GHG Cost ($/tCO_2e avoided) for diesel truck M-TES varies with the distance. Results are shown for the levelized cost estimation of M-TES with a system level ESD of 0.7 MJ/kg and other Surrey DEN low-carbon sources.

## 5. Conclusions

Industrial waste heat sources are abundant and underutilized because of the typical long distance between industrial locations and the demand such as district energy networks. To overcome the distance limitation, a proposed system of mobile thermal energy storage can be used. To use the M-TES system in the SCE, it is essential to assess economic feasibility. The economic and environmental costs are determined based on \$/MWh and tCO2e/MWh of M-TES considering both the heat demands and heat transport distances. M-TES has also been compared with other low carbon sources that the City is currently considering. M-TES can be used for delivering waste heat to district energy networks in urban centers. Based on the techno-economic and GHG reduction of multiple M-TES configurations, M-TES showed promising results when compared to other low-carbon energy sources, such as biomass energy and sewer heat recovery. The value of such thermal storage systems depended not only on the system-level ESD but also on the distance of the waste heat source from the DEN, schedule, and transportation mode.

- With a fixed schedule of six trips/day and 360 days/year, an M-TES truck with a capacity of 10 tonnes and system level ESD of 0.7 MJ/kg can meet up to 7% of the SCE network anticipated demand for 2022.
- The levelized cost of energy of M-TES increases with distance and decreases with the energy storage density of the selected solution.
- The most efficient configuration of M-TES is achieved with the highest ESD and shortest distance between the industrial heat source and the DEN.
- Out of the three truck modes to be used with M-TES, the electric vehicles (EV) truck is slightly more competitive than the renewable natural gas (RNG) truck in BC. However, both RNG and electric trucks are better than diesel trucks in cost and GHG-avoided totals.
- As a result, when using RNG or electric trucks to move the tank between the IWH and DEN, M-TES is competitive with a distance range of 15–50 km.
- Diesel trucks are not efficient if the distance is more than 30 km.

**Author Contributions:** Conceptualization, M.S. and M.B.; Methodology, M.S.; Writing—Original Draft Preparation, M.S.; Review & Editing, E.K., J.O. and M.B.; Supervision, M.B.; Formal analysis, M.B. and M.S. All authors have read and agreed to the published version of the manuscript.

**Funding:** The authors declare no conflict of interest.

**Institutional Review Board Statement:** Not applicable.

**Informed Consent Statement:** Not applicable.

**Data Availability Statement:** Restrictions apply to the availability of these data. Some data was obtained from City of Surrey and are available from the authors with the permission of City of Surrey.

**Acknowledgments:** This research is supported by funding from the Pacific Institute for Climate Solutions (PICS) Opportunity (Grant No. 36170-50280) and NSERC Advancing Climate Change Science in Canada (Grant No. 536076-18). Financial support and valuable technical insights were provided by E. Kwok, D. Moore, and J. Owen of the City of Surrey.

**Conflicts of Interest:** The authors declare no conflict of interest.

## Abbreviations

| | |
|---|---|
| CC | Capital Cost (\$) |
| BAU | Business as usual |
| D | Distance between source and sink (km) |
| DEN | District Energy Network |
| ESD | Energy storage density (MJ/kg) |
| EV | Electrical Vehicle |

| FP | Fuel price |
|---|---|
| GHG | Greenhouse gases |
| $GHG_{Transportation}$ | Total GHG emitted from the system ($tCO_2e$/ year) |
| IWH | Industrial waste heat |
| LCM | Levelized cost of M-TES ($/MWh) |
| M-TES | Mobile thermal energy storage |
| N | Number of trips per day |
| ODPY | Number of operational days per year |
| OTC | Other transportation cost (insurance + maintenance + Driver rate) ($/km) |
| PCM | Phase Change Materials |
| RNGV | Renewable natural gas-powered vehicles |
| SCE | Surrey City Energy |
| SLT | System lifetime (years) |
| TC | Total Transportation Cost ($) |
| TCM | Thermochemical materials |
| TEA | Techno-Economic Analysis |
| TH | Total heat provided by the system per year |
| TW | Total material weight (kg) |
| $\eta_{Fuel}$ | Fuel efficiency in (L/100 km) or (kWh/100 km) |
| $\eta_{System}$ | M-TES thermal efficiency |

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
