# Peer review of "Integrating Mobile Thermal Energy Storage (M-TES) in the City of Surrey’s District Energy Network: A Techno-Economic Analysis"

_applsci, doi:10.3390/app11031279_

Round 1

Reviewer 1 Report

REVIEW:

The manuscript “Integrating Mobile Thermal Energy Storage (M-TES) in the City of Surrey’s District Energy Network: A Techno-economic Analysis” sent to Applied Sciences, presents an technoeconomic analysis of the possibility to utilize industrial waste heat in a district heating (DH) system of a city by transferring this waste heat with trucks. The levelized cost of energy and cost of avoided CO2 emissions are calculated for M-TES with varying Energy Storage Density (EDS), distance of travel and with varying truck types. These values are compared to the existing way of producing DH (a natural gas boiler) and to other renewable heat sources. The results show that the M-TES technology utilizing industrial waste heat can be competitive.

Developing knowledge and technology for increasing the share of renewable energy and heat in cities is an extremely important and timely research topic and hence the research topic of this manuscript is important and suitable to be presented in Applied Sciences.

The manuscript has been decently well written language-wise. The structure is clear. The motivation and objective of the manuscript is clearly presented and to some extent also the methodology. The diagrams of the results are decently clear, but the results are not discussed basically at all. The conclusions are based on the results. The problem is in the sloppy style of writing. In my printed version many references are missing, some abbreviations are not presented, and some data not provided.  

I feel that the novelty of this manuscript is enough for publishing in Applied Sciences. Certainly not any fundamental scientific or even technological novelty exists, but the technoeconomic analysis provides valuable new information in the opportunity to utilize M-TES with industrial waste heat.

There is a need for Major Revision due to the sloppy writing, but generally the manuscript has potential to be published in Applied Sciences especially if the following comments/questions are addressed properly:

Comments/questions:

  • There are many times a text “Error! Reference source not found”. Correct these.
  • Why did you not compare M-TES with the possibility to invest in a pipeline?
  • The results need more discussion
  • Why is in Figure 6 the levelized cost of diesel truck with 15 km less than 40 $, but in Figure 7 the value is 42 $? Same happens with avoided GHG costs.
  • You could indicate that you (probably) are using Canadian dollars.
  • You could indicate which TES materials can reach the presented values of different ESD, because now these are a bit in air and in your analysis these can reach values up to 2 MJ/kg (Figure 5) or at least describe what means System level ESD.
  • I don’t understand equation 6 and the misleading explanations below it doesn’t help.
  • FP is missing in Nomenclature
  • The values of FP are not given.
  • In the Nomenclature you have RNGV (Renewable natural gas-powered vehicle). Can you explain me what do you mean with renewable natural gas? Bio-based methane??
  • It can be concluded based on your results that the levelized cost of energy of M-TES increases with the distance to the industrial waste heat site, but I am not sure is this conclusion such that it needs to be included in the Conclusions

FINAL RECOMMENDATION from REVIEWER: MAJOR REVISION

Reviewer 2 Report

In this paper, a techno-economic analysis study on integrating mobile thermal energy in the city of Surrey’s district energy network. The energy storage density of the proposed mobile thermal energy storage is determined. In terms of mobile thermal energy storage, what are the CO2 emission implications as well as cost implications? In addition, see below further suggestions for improvement

  1. The abstract is not clear about what low carbon energy sources are being considered
  2. Throughout the paper, leave spaces between the numerical values and the units. i.e. 30 km and not 30km see lines 11 and 12.
  3. There is a number of reference errors that should be sorted out to help with continuity of the study see line 65, 97, 149.
  4. The introduction is very weak with no extensive review and citation of related literature. There might not be studies on mobile thermal energy storage, but there is literature on district energy networks. The issues with these networks that necessitate mobile thermal energy storage should pointed out.
  5. Section 2.1, there is need for detailed description of the SCE and possibly an idea of where the buildings are and how far they are from each other. Is mobile thermal energy storage the best option for such building? How big are the buildings, can onsite production of heat be a better alternative in the long run?
  6. Breakdown the heat from low carbon sources, since these sources form a big percentage of the annual demand in figure 2, does it make sense using these low carbon energy sources for onsite generation of heat? I.e. Solar thermal can be used onsite, geothermal can be used on site.
  7. Depending on the source of waste heat since you are considering transportation of this waste heat, what is the amount that is needed for the demand being considered in the study? i.e. How many cement plants are needed to supply all the demand?
  8. I think the study lacks a solid technical merit, the determination of energy storage should be backed by clear theory and corresponding clear assumptions. Section 3.2 is presented without consideration of where the waste heat is coming from and how much this is. Earlier, the authors mentioned the source of heat as cement factories and waste from sewer lines. These are missing in the analysis and the feasibility of such a system depends on the availability of such and the costs involved.
  9. In the results, what would be the comparison with case of onsite generation? Moreover, there are lots of missing references which make this section to read.
  10. In general, the analysis and the results themselves are shallow and need to be considerably improved. Authors should talk about onsite generation and storage and compare this with mobile thermal energy storage.

Round 2

Reviewer 1 Report

My comments/questions have been addressed decently well, at least in the manuscript, so my suggestion is to ACCEPT this manuscript. But as a a remark it has to be said that in the response letter it should be said where and how and why/why not the change has been. It is not enough to say "I can do this".

Reviewer 2 Report

The authors did not answer the raised queries satisfactorily, although in some cases the reviewers say changes could be made, these changes have not been made in the manuscript.  

Round 3

Reviewer 2 Report

The authors have addressed the raised queries satisfactorily